# Speech Codecs Beyond Compression: Towards Autoregressive Generative Modeling

## Abstract

Recent advances in speech language models have leveraged discrete speech representations from pretrained codecs to enable scalable training and generation. However, existing codecs are primarily designed for compression, without accounting for the autoregressive nature of language model training. This mismatch leads to suboptimal performance when using compressed speech tokens for sequence modeling. In this work, we revisit speech discretization from the perspective of generative modeling and propose a novel framework to align tokenization with the autoregressive training paradigm. Specifically, we introduce autoregressive-compatible constraints into the codec training process, encouraging token sequences that better reflect the temporal consistency and predictability expected by language models. In addition, we propose using heterogeneous sampling strategy for different layers of audio tokens (semantic versus acoustic) to enhance the alignment between semantic tokens and the speech's textual content. Experiments across multiple benchmarks demonstrate that our approach bridges the gap between audio compression and generative modeling, enabling more effective continued pretraining of existing large language models on audio data. Consistent performance gains across multiple codecs further validate the generalizability of our proposed method.

## 1 Introduction

Large language models (LLMs) have achieved remarkable success across a wide range of natural language processing tasks (Zhao et al., 2023; Yang et al., 2024; Team, 2025), and recent efforts have extended their capabilities to audio by incorporating discrete representations extracted from audio codecs (Du et al., 2023; Xie & Wu, 2024; Wu et al., 2024; Veluri et al., 2024; Défossez et al., 2024). These codecs, such as XCodec (Ye et al., 2025a), use residual vector quantization (RVQ) (Zeghidour et al., 2021; Défossez et al., 2022) with multiple codebooks, and encode the waveform into a set of integer token streams, each representing a quantized latent sequence. At each time step, there are multiple integers, one per codebook, forming a richer representation. By mapping tokens back to their codebook embeddings and combining across multiple codebooks, one can recover the approximate latent representation of the original audio, which the decoder then converts back into waveform. The generated discrete tokens enable LLMs to process and generate audio using the same autoregressive modeling paradigm as in text.

However, while such codecs are highly effective for compression efficiency and perceptual quality, their tokenization schemes are not explicitly designed with autoregressive modeling. This misalignment raises a critical issue: current audio tokenizers produce discrete sequences without modeling the conditional dependencies that autoregressive LLMs assume by design. As a result, token combinations are not necessarily consistent with the next-token prediction paradigm, weakening the temporal consistency and predictability of audio token streams. To further illustrate this discrepancy, we examine the statistical distributions of audio tokens generated by several representative codecs. As shown in Figure 1, unlike textual tokens that naturally follow a Zipfian distribution (Kingsley Zipf, 1932), audio tokens deviate significantly from this pattern, even at the unigram level. This divergence highlights the fact that current compression-oriented tokenizers do not capture the structured frequency-rank relationships that autoregressive models rely on, thereby increasing the learning burden during multimodal training. Addressing this gap forms a key motivation for our work.

In this work, we revisit the problem of speech tokenization from the perspective of generative modeling and propose a novel codec training framework that explicitly encourages autoregressive compatibility. Our method introduces an autoregressive regularizer, which incorporates an auxiliary next-token prediction loss via an autoregressive decoder during codec training. This constraint aligns the structure of the compressed speech tokens with the learning dynamics of LLMs, ensuring that token sequences are both compact and autoregressively coherent. While recent speech codecs often adopt multi-layer token hierarchies to reduce the length of generated token sequences, the overall token rate remains significantly higher than that of the corresponding text content, posing challenges for alignment with language models. Therefore, the top layer of audio tokens, that typically represents seman-

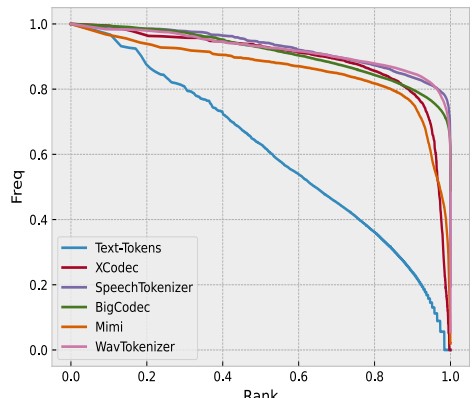

Figure 1: Zipf's Law observed in various representative audio codecs, illustrating the power-law relationship between token frequency and rank commonly seen in natural language.

tic rather than acoustic information, is especially amenable to further compression. Motivated by this, we introduce a heterogeneous sampling strategy that applies a lower sampling rate to semantic token layer and higher rates to acoustic ones. In our experiments, we successfully reduce the final frame rate to 6.25Hz, which shortens the sequence length and improves autoregressive modeling efficiency. By compressing the semantic layer to a token rate closer to that of the transcript, we achieve better alignment with the textual modality, making the resulting tokenizations more suitable for integration with LLMs in cross-modal tasks.

Our approach is model-agnostic and can be integrated into various existing speech codecs. We demonstrate its generality by applying it to multiple representative codecs and validating its effectiveness through comprehensive experiments on audio generation tasks. Results show that our method not only maintains strong compression quality but also significantly enhances the performance of LLMs in audio modeling, paving the way for more effective large-scale audio-language training. Our contributions are as follows:

- We identify and analyze the mismatch between conventional speech codecs and the autoregressive nature of LLMs, highlighting the combinatorial ambiguity in audio tokenization as a key challenge for cross-modal modeling.

- We propose an autoregressive-oriented codec training framework that introduces a next-token prediction regularizer, which biases the quantizer toward producing autoregressively coherent token sequences. Furthermore, we incorporate a heterogeneous sampling strategy across token hierarchy levels to reduce semantic token rates and better align them with textual data, thereby enhancing efficiency and cross-modal compatibility.

- Our approach is generalizable to various audio codecs. We demonstrate its effectiveness in multiple representative codecs and show consistent improvements across generation quality, statistical properties of tokens, and downstream LLM training and tasks without sacrificing compression and tokenization ability.

## 2 RELATED WORK

**Audio Codecs and Discrete Representations** Recent progress in neural audio codecs has enabled efficient compression of waveform signals into discrete token sequences (Zhang et al., 2023a; Défossez et al., 2024; Wu et al., 2024; Ye et al., 2025a). Models such as Encodec (Défossez et al., 2022), XCodec (Ye et al., 2025a), CodecBPE (Shen et al., 2024), and FunCodec (Du et al., 2024) adopt vector quantization Zeghidour et al. (2021) and multi-level encoders to capture high-fidelity audio representations. These systems are typically optimized for rate-distortion trade-offs and perceptual metrics (e.g., PESQ (Rix et al., 2001), UTMOS (Saeki et al., 2022)), but are not designed to support the sequential dependencies required by autoregressive models. Although several codecs (Zeghidour et al., 2021; Kumar et al., 2023; Ye et al., 2025a) adopt causal architectures

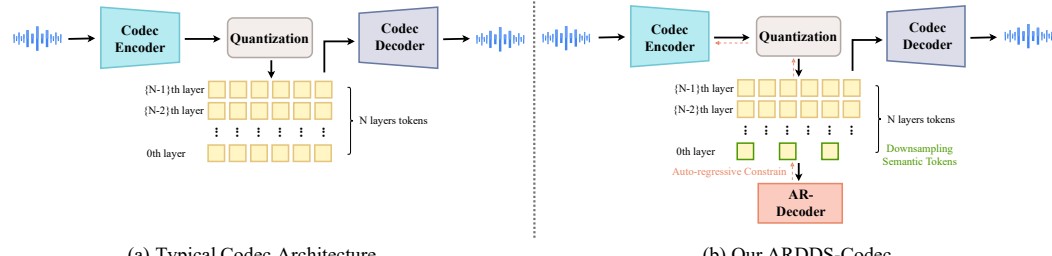

(a) Typical Codec Architecture                    (b) Our ARDDS-Codec

Figure 2: **(a)** A general architecture of audio codecs, which convert raw audio into hierarchical token representations across multiple layers. **(b)** Overview of our method, that integrates standard audio codecs with autoregressive regularization and heterogeneous downsampling of the first-layer tokens.

that prevent information leakage from future tokens into past hidden states, such architectures do not necessarily guarantee the autoregressive predictability of the generated speech tokens. This limitation is further evidenced by the statistical analysis presented in Figure 1. More recent works like Moshi (Défossez et al., 2024) build on these representations for audio generative modeling, but majorly rely on multiple layer audio tokens organized with delay pattern. Our work diverges by modifying the codec training objective itself to make the resulting tokens inherently autoregressive-compatible.

**Speech-Language Modeling with LLMs** Efforts to extend LLMs to audio include models like Llama-Omni (Fang et al., 2024), SpeechGPT (Zhang et al., 2024), and SPIRIT-LM (Nguyen et al., 2025), which typically process either continuous features or discrete tokens produced by codecs. While some models fine-tune pretrained encoders or align speech with text via contrastive learning (e.g., CLAP (Elizalde et al., 2022), BEATs (Chen et al., 2023)), recent trends focus on representing audio as a language (e.g., GLM-4-Voice (Zeng et al., 2024)) to reuse the autoregressive training paradigm within LLM through treating audio tokens as textual tokens. However, these approaches inherit the limitations of their underlying audio tokenizers, which are not optimized for next-token predictability. Our work addresses this limitation directly at the token generation level by regularizing the codec to produce LLM-adaptable token sequences.

**Autoregressive Modeling & Tokenization Constraints** In the text domain, previous works have explored the importance of tokenizer alignment for generation quality, such as token merging (Xu et al., 2022), vocabulary regularization (Xu et al., 2021), and prefix tuning (Li & Liang, 2021). In vision and speech, efforts like VQGAN (Esser et al., 2021) and SpeechTokenizer (Zhang et al., 2023a) explore more expressive token spaces, but typically ignore autoregressive compatibility. In contrast, our approach introduces an autoregressive decoder into the codec training loop, enforcing next-token predictability as an inductive bias. This improves token consistency, and bridges the gap between compression and generative modeling, a direction that remains underexplored in the current literature.

## 3 METHODOLOGY

We propose a novel training framework for audio codecs that explicitly aligns the generated discrete token sequences with the autoregressive learning paradigm of large language models (LLMs), as shown in Figure 2. Our key idea is to incorporate an autoregressive regularization objective directly into the codec training process, ensuring that the resulting token sequences are both compressive and autoregressively predictable. This section presents the key components of our proposed methodology. We begin with a brief overview of standard audio codecs (§3.1), followed by a description of our autoregressive regularization approach (§3.2). Lastly, we introduce our heterogeneous downsampling strategy (§3.3).

### 3.1 PRELIMINARIES

Let $x = [x_1, x_2, \ldots, x_T]$ denote a raw audio waveform. A conventional neural audio codec typically consists of three components: an encoder, a quantizer, and a decoder.

**Codec Encoder** The encoder $\mathcal{E}_\theta$ maps the continuous audio signal into a sequence of latent features:

$$z = \mathcal{E}_\theta(x) = [z_1, z_2, \ldots, z_N] \tag{1}$$

where $z_i \in \mathbb{R}^d$ and the length of audio tokens $N \ll T$ due to downsampling.

**Quantization to Discrete Tokens** A central component in neural audio codecs is the quantization module $\mathcal{Q}$, which discretizes latent features into integers. The latent vectors $z$ are discretized by a vector quantization module (e.g., residual vector quantization (RVQ) or product quantization), producing discrete token indices:

$$y = \mathcal{Q}(z) = [y_1, y_2, \ldots, y_N], \quad y_i \in \{1, 2, \ldots, K\} \tag{2}$$

where $K$ is the size of the codebook. Depending on the codec, each $z_i$ may be quantized into one or more codebook levels. In this work, we focus on multi-level audio tokens as our heterogeneous downsampling strategy treats acoustic and non-acoustic tokens with different sampling rates.

In addition, another quantization method known as Finite Scalar Quantization (FSQ) has been attracting increasing attention. In FSQ, each dimension of a latent vector is quantized independently into a fixed number of bins.

**Codec Decoder** The decoder $\mathcal{D}_\phi$ reconstructs the audio waveform from the discrete tokens:

$$\hat{x} = \mathcal{D}_\phi(y) \tag{3}$$

The encoder $\mathcal{E}_\theta$ and decoder $\mathcal{D}_\phi$ are typically implemented using convolutional neural networks (CNNs) to exploit the local structure of audio signals efficiently.

**Training Objective** The traditional codec is trained based on the GAN framework using multiple losses: 1) *The reconstruction loss*: A multi-scale Mel-spectrogram reconstruction loss, computed as the L1 distance in the spectral domain across multiple scales. The Mel-spectrogram serves as a perceptually relevant representation of audio and is closely correlated with human auditory perception.

$$\mathcal{L}_{\text{recon}} = \|x - \hat{x}\|^2 \tag{4}$$

2) *Least-square GAN loss*: Two types of discriminators are typically employed during codec training: Multi-Period Discriminator (MPD) (Kong et al., 2020) to capture pitch-dependent periodicity, and the Multi-Scale STFT Discriminator (MS-STFT) (Défossez et al., 2022), which operates in the spectral domain to assess fidelity across resolutions. 3) *Discriminator Feature Loss*: Also known as perceptual loss (Ledig et al., 2017), this L1 feature-matching objective encourages the generator to produce perceptually natural outputs by aligning intermediate discriminator activations. 4) *Quantization Loss*: For vector quantization, the L1 loss is applied to train the codebook, enabling bidirectional conversion between continuous features and discrete audio tokens.

This standard training paradigm focuses solely on reconstruction quality and perceptual fidelity, with no constraints on the sequential structure of the token sequence $y$. As a result, the resulting tokens may not exhibit the kind of predictable, conditionally dependent patterns required by autoregressive language models. This limitation motivates our proposed method, which augments the codec training with an autoregressive regularization objective.

## 3.2 Autoregressive Regularization

To ensure that the token sequence $y$ is more compatible with autoregressive modeling, we introduce an auxiliary decoder $\mathcal{A}_\psi$ trained to predict the next token in the sequence, thereby encouraging the codec to produce token sequences that follow the autoregressive paradigm:

$$\mathcal{L}_{\text{AR}} = -\sum_{t=1}^{N-1} \log P(y_{t+1}|y_{\leq t}; \psi) \tag{5}$$

where $\mathcal{A}_\psi$ can be a lightweight Transformer-based decoder or RNN-based decoder. It is jointly optimized with the codec components.

A critical challenge in applying this autoregressive regularization is that discrete token IDs $y$ are non-differentiable, making it difficult to backpropagate gradients through the quantization module. To address this, we adopt a soft approximation of the discrete token assignment. Let $x_t^{\text{feat}}$ be the

output feature from the encoder at timestep $t$, and let $\mathcal{C} = \{c_1, c_2, \ldots, c_K\}$ denote the codebook. Instead of directly using the hard argmin operation to select the nearest codeword, we compute a soft assignment probability using a temperature-controlled softmax over negative distances:

$$p_t(i) = \frac{\exp\left(-\|x_t^{\text{feat}} - c_i\|^2/\tau\right)}{\sum_{j=1}^{K} \exp\left(-\|x_t^{\text{feat}} - c_j\|^2/\tau\right)} \tag{6}$$

where $\tau$ is a temperature parameter. As $\tau \to 0$, $p_t(i)$ becomes a sharp distribution approximating one-hot, and the soft assignment converges to the argmin selection. This soft token distribution is then used as input to the autoregressive decoder $\mathcal{A}_\psi$, enabling gradient flow back through the quantization module and the encoder. This design allows the codec to produce discrete token sequences that not only support compression quality but also align well with autoregressive modeling.

The overall training objective is a combination of the original codec loss $\mathcal{L}_{\text{ori}}$ and the regularization:

$$\mathcal{L} = \mathcal{L}_{\text{ori}} + \lambda \mathcal{L}_{\text{AR}} \tag{7}$$

where $\lambda$ is a weighting coefficient that balances the trade-off between compression fidelity and autoregressive predictability.

This joint training framework encourages the codec to produce token sequences that are both semantically meaningful for reconstruction and structurally aligned with next-token prediction—a critical property for continued pretraining in LLMs.

### 3.3 Heterogeneous Downsampling

While audio codecs have adopted multi-layer token representations to reduce sequence length, a significant mismatch still remains between the number of audio tokens and the number of textual tokens typically used in language modeling. This discrepancy poses challenges when aligning audio with text in multimodal large language models. Considering a speech audio, there is potential to compress the semantic tokens to a length that more closely matches the number of text tokens in its transcript. Notably, the 0th layers tokens often contain content information that is more relevant to language modeling than acoustic layers, making them ideal candidates for further compression.

To address this issue, we propose a *heterogeneous downsampling* strategy that selectively reduces the token rate of 0th quantization layer more aggressively than those of acoustic layers. Formally, for a codec producing $L$ token streams $\{y^{(0)}, y^{(1)}, \ldots, y^{(L-1)}\}$ from bottom (semantic) to top (acoustic), we assign downsampling rates $\{r_0, r_1, \ldots, r_{L-1}\}$ such that $r_0 < r_1 = r_2 = \ldots r_{L-1}$. This allows 0th-layer tokens to appear at a coarser temporal resolution, aligning their frequency more closely with that of textual tokens, while preserving the finer granularity of acoustic details at lower levels. In our implementation, we apply downsampling at the feature level during quantization process. Specifically, for the first layer, we perform average pooling over the latent feature vectors within a fixed window size $W_{\text{ds}}$, effectively summarizing coarse information. This reduces the number of token emissions while maintaining representational fidelity.

By compressing the first layer to a lower token rate while maintaining higher rates for acoustic layers, our design enhances the alignment between the semantic content of audio and textual modalities. This results in audio token sequences that are more compatible with the autoregressive modeling paradigm used in large language models (LLMs). Our strategy is also motivated by recent Audio LLM frameworks (Le Lan et al., 2023; Wang et al., 2023; Défossez et al., 2024), which adopt a delayed generation mechanism wherein textual tokens are generated prior to the corresponding audio tokens—a paradigm known as text-guided audio generation. This approach has been shown to outperform purely audio-driven generation. When semantic audio tokens appear at a temporal resolution comparable to that of text tokens, the interleaved generation becomes more coherent and structurally aligned, thereby improving both generation quality and the integration between text and audio modalities. Heterogeneous downsampling complements our autoregressive regularization (Sec. 3.2), jointly encouraging the codec to produce audio tokens that are compact, expressive, and structurally aligned with the learning paradigm of LLMs. Our framework is modular and can be integrated with various vector-quantization-based codecs, including XCodec, SpeechTokenizer, and BigCodec variants. This ensures high generalizability across different model families and use cases.

Table 1: Downstream performance of SpeechLM trained with different audio codecs. Evaluation metrics include accuracy (%) for StoryCloze and TopicStoryCloze, character error rate (CER ↓) for AISHELL-I (ASR), and word error rate (WER ↓) for SeedTTS (TTS).

| Method / Task | StoryCloze↑ | TopicStoryCloze↑ | AIShell-I↓ | SeedTTS↓ |
|---|---|---|---|---|
| XCodec (Ye et al., 2025a) | 63.7 | 72.3 | 5.2 | 4.7 |
| SpeechTokenizer (Zhang et al., 2023a) | 63.4 | 72.5 | 4.9 | 4.6 |
| XCodec-2 (Ye et al., 2025b) | 64.2 | 72.8 | 4.7 | 4.2 |
| BigCodec Xin et al. (2024) | 63.9 | 71.7 | 5.3 | 4.9 |
| ARDDS-XCodec(Ours) | **70.1** +6.4 | **76.9** +4.6 | **2.5** -2.7 | **2.3** -2.4 |

## 4 EXPERIMENTS & ANALYSIS

We conduct comprehensive experiments to evaluate the effectiveness of our proposed autoregressive-compatible codec training framework. Our evaluation spans multiple codec architectures, statistical properties of generated token sequences, and end-to-end training of speech language models (SpeechLM) following with downstream tasks' evaluation.

### 4.1 EXPERIMENTAL SETTINGS

**Codec** To ensure fair comparisons across different codecs, we train all models on the same LibriSpeech 960-hour corpus. Specifically, we adopt the standard 585-hour clean subset for codec training, which is widely used in previous work (Zhang et al., 2023a; Du et al., 2024; Ye et al., 2025a). All audio is resampled to 16 kHz. We evaluate our method on several representative codecs whose training implementations are publicly available.

For each codec, we retain the original architecture and training hyperparameters, but standardize the training data and sampling rate across all experiments. For our autoregressive-compatible training, we set the softmax temperature in Eq.(6) to 0.01 after a grid search. This value results in a distribution that closely approximates one-hot behavior, without yielding further gains at lower temperatures in our experiments. The regularization weight $\lambda$ in Eq.(7) is set to 1. We reduce the final frame rate of the first quantization layer to 6.25Hz, thereby shortening the sequence length. However, this configuration already represents the stability limit in our experiments; further reductions in the frame rate result in training instability, even after extensive hyperparameter tuning. We hypothesize that this instability arises because reducing the frame rate below 6.25Hz (e.g. 3Hz) yields fewer speech tokens per second than textual tokens in the corresponding transcript under XCodec's setting. Our autoregressive decoder is a lightweight Transformer decoder, with 6 layers, a hidden size of 2048, and 16 attention heads. It is jointly trained with the codec encoder and quantization module. We use XCodec (Ye et al., 2025a)[1] as the representative multi-layer residual-vector-quantization (RVQ) codec to conduct our experiments.

**SpeechLM** To evaluate the downstream effectiveness of our method, we perform large-scale speech-language modeling (SpeechLM) by continuing pretraining on a modified LLaMA-3 8B model. We make minimal architectural changes: (1) adding learnable embeddings for audio tokens, and (2) introducing a separate LM head for audio. Audio tokens are first embedded and then concatenated with textual tokens for autoregressive decoding. We follow prior works (Xie & Wu, 2024; Défossez et al., 2024; Ye et al., 2025a) in employing a delay-based generation strategy for multi-layer audio tokens. Training data consists of a large-scale Chinese audio corpus containing approximately 400k hours. All speech samples are tokenized with the trained codecs under evaluation.

We design three tasks for training: automatic speech recognition (ASR), text-to-speech (TTS), and interleaved text-audio modeling. The speech-text interleaving pattern segments a long-form speech utterance into multiple blocks. For each block, either audio tokens or the corresponding textual tokens from the transcript are randomly selected as the representation. This mixed-modality sequence is then used as input to the speech-language model, encouraging the model to learn fine-grained cross-modal alignment and flexible generation capabilities. This speech-text interleaving approach has also been adopted in prior work to facilitate multi-modal alignment and generation between audio and text (Zhang et al., 2023b; Défossez et al., 2024; Xie & Wu, 2024). In practice, we find

---

[1]https://github.com/zhenye234/xcodec

Table 2: Ablation results of different copponents proposed in our method. The left part demonstrates the downstream performance of SpeechLM trained with different audio codecs, while the right part shows the analysis of codecs' original compression ability.

| Method / Task | StoryCloze↑ | TopicStoryCloze↑ | AIShell-I↓ | SeedTTS↓ | WER↓ | SPKSIM↑ | UTMOS↑ |
|---|---|---|---|---|---|---|---|
| Ours | 70.1 | 76.9 | 2.5 | 2.3 | 4.13 | 0.88 | 4.30 |
| w/o AR | 67.0 | 74.2 | 4.2 | 3.5 | 4.11 | 0.90 | 4.25 |
| w/o DDS | 67.1 | 74.7 | 3.8 | 3.4 | 4.14 | 0.90 | 4.27 |
| w/o both | 63.7 | 72.3 | 5.2 | 4.7 | 4.10 | 0.86 | 4.24 |

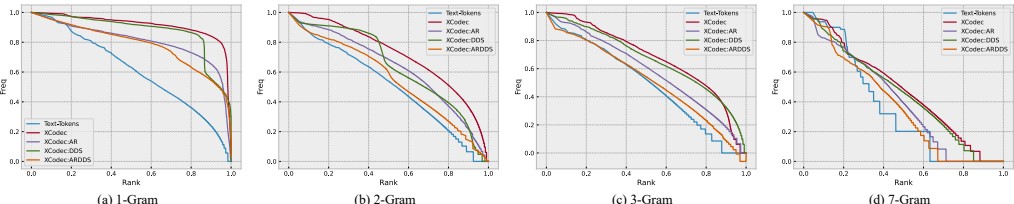

(a) 1-Gram  (b) 2-Gram  (c) 3-Gram  (d) 7-Gram

Figure 3: Zipf's Law analysis (normalized token log-frequency against normalized Log-Rank for several audio and textual languages) on 1-gram, 2-gram, 3-gram, and 7-gram token frequency distributions. "XCodec:AR" denotes XCodec with only autoregressive regularization, "XCodec:DDS" applies only heterogeneous downsampling strategy, and "XCodec:ARDDS" is our full method.

that ASR and TTS are easier to optimize, while interleaved modeling better reflects the complexity of cross-modal generation. Following Zeng et al. (2024), we adopt the same dynamic sampling ratio of 90:1:1 for interleave, ASR, and TTS tasks, respectively. The Audio LLM is trained using the Megatron-LM framework[2] on 32 NVIDIA A100 GPUs. We use Adam as the optimizer with $\beta_1 = 0.9$, $\beta_2 = 0.95$, and $\epsilon = 10^{-8}$, and an initial learning rate of $1e-4$. Training continues until convergence on held-out validation sets.

## 4.2 SPEECHLM PERFORMANCE

We first experiment with XCodec, and compare performance with other open-sourced baselines. To assess whether our codec outputs are better suited for autoregressive modeling, we train speechLLM upon audio tokens tokenized from the audio dataset introduced on § 4.1 with different codecs.

**Overall results** We begin by evaluating the effectiveness of our proposed codec training framework in enabling better SpeechLM. As shown in Table 1, SpeechLM trained on audio tokens from our ARDDS-XCodec achieves substantial gains across four downstream tasks—including two speech-language reasoning benchmarks (StoryCloze, TopicStoryCloze) and two generation-based speech tasks (AIShell-I for ASR and SeedTTS for TTS). These results consistently confirm that the audio tokens produced by our codec training framework are more suitable for autoregressive modeling in SpeechLM. The improvements validate that enforcing autoregressive compatibility during codec training leads to token sequences that better align with the next-token prediction dynamics required by LLMs, thereby improving overall speech-language modeling quality.

**Impact to codec's original performance** A natural concern is whether such performance gains come at the cost of degraded codec performance. As shown in Table 2 (right), our augmented codecs ("Ours") maintain comparable scores to the vanilla baseline ("w/o both") in standard metrics for audio reconstruction quality (WER), speaker similarity (SPKSIM), and perceptual naturalness (UTMOS). This confirms that our method does not harm the core capabilities of the codec, preserving its compressive and perceptual performance.

## 4.3 WHY OUR METHOD WORKS

**The language of audio tokens** To further understand why our method improves autoregressive modeling, we analyze the statistical properties of the audio token sequences using Zipf's Law. In natural language, token frequencies typically follow a power-law distribution, known as Zipf's Law, where the frequency of a token is inversely proportional to its rank. This statistical regularity reflects

---

[2]https://github.com/NVIDIA/Megatron-LM

the structured and compressible nature of language and has been widely used as an indicator of how well a token sequence is suited for autoregressive modeling and next-token prediction.

We adopt this perspective to examine whether audio tokens produced by different codecs exhibit similar Zipfian behavior. As illustrated in Figure 3, token distributions from our proposed ARDDS-XCodec show a significantly more Zipf-like pattern across 1-gram to 7-gram statistics compared to baselines. This trend suggests that our autoregressive regularization effectively reshapes the token distribution to be more structurally aligned with natural language, thereby improving the predictability and learnability of the sequence for autoregressive LLMs.

**SpeechLM Training Efficiency** As shown in Figure 4, models trained with our ARDDS-XCodec tokenizer exhibit clearly faster convergence compared to those trained with the vanilla XCodec. The loss curve of ARDDS-XCodec drops more rapidly in the early stages and stabilizes at a lower final loss, indicating improved learning efficiency and better compatibility of the generated audio tokens with autoregressive modeling.

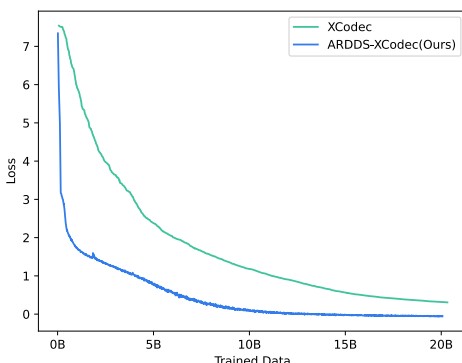

Figure 4: SpeechLM's training loss vs. trained data volume using XCodec and ARDDS-XCodec.

## 4.4 ABLATION STUDY & ANALYSIS

To better understand the contribution of each component in our proposed codec training framework, we conduct ablation studies based on the XCodec model. Specifically, we evaluate three variants: one without the autoregressive regularizer (w/o AR), one without heterogeneous downsampling (w/o DDS), and one with both components removed (w/o both). The results are shown in Table 2, which reports both downstream SpeechLLM performance (left) and intrinsic codec evaluation metrics (right).

**Ablation analysis of different components** As shown in Table 2 (left), removing either component results in consistent degradation across all downstream tasks, including reasoning (StoryCloze), speech understanding (AISHELL-I), and speech generation (SeedTTS). The full model (ARDDS-XCodec) achieves the best overall performance, while removing both components leads to results of the original XCodec baseline. These findings confirm that the observed gains stem from the combined effect of autoregressive regularization and heterogeneous downsampling.

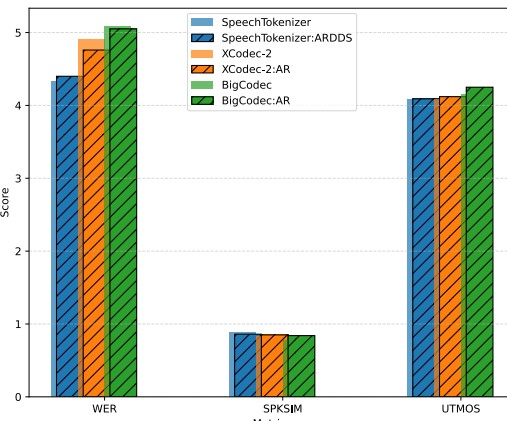

Figure 5: Codec quality comparison. Our method augmented codecs do not degenerate the performance against their vanilla ones.

**Impact to Codecs' original performance** Importantly, none of the variants exhibit significant degradation in codec quality. Metrics such as WER, SPKSIM, and UTMOS remain comparable across configurations, indicating that our proposed modifications do not compromise the original reconstruction quality or speaker characteristics of the codec.

**Zipfian's law analysis** To further support this observation, we examine the statistical structure of the generated token sequences. As shown in Figure 3, which visualizes the $n$-gram token frequency distributions, our method (and its partial variants) consistently shifts the token statistics closer to the Zipfian distribution observed in natural language. The more Zipfian distribution of audio tokens indicates reduced redundancy and higher predictability, which in turn facilitates more efficient learning under the autoregressive modeling paradigm. These results suggest that our approach guides the codec toward generating more autoregressive-compatible token sequences, without degenerating compression quality.

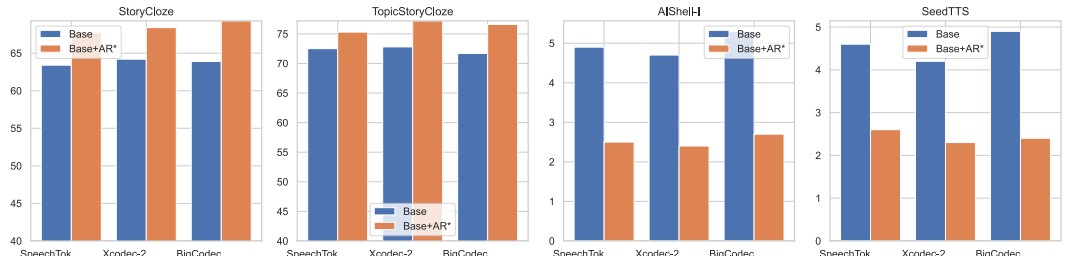

Figure 7: Performance comparison between baselines and our method augmented counterparts across multiple evaluation tasks for the trained SpeechLM.

**Generalization Analysis** To evaluate the generality of our proposed autoregressive-compatible codec training framework, we apply it to three additional representative codecs: `SpeechTokenizer`, `BigCodec`, and `XCodec-2`. Among them, `SpeechTokenizer` and `BigCodec` are another strong open-source baselines, while `XCodec-2` is an upgraded version of XCodec that employs a single-layer finite scalar quantization (FSQ) scheme for improved simplicity and efficiency. For these codecs, we augment them with our autoregressive regularization (denoted as `XXX:AR`) while keeping the codec architecture and reconstruction objective unchanged. Since `XCodec-2` and `BigCodec` are single-layer models, we only apply the autoregressive regularization without heterogeneous downsampling. As indicated by the WER, SPKSIM, and UTMOS scores shown in Figure 5, our modifications do not degrade the original compression performance or perceptual quality of the codecs. The augmented codecs retain comparable reconstruction ability, while enabling better alignment with the next-token prediction paradigm required by LLMs.

We further visualize the Zipf's Law statistics (1-gram frequency vs. rank) for each codec in Figure 6. The curves of our enhanced codecs shift closer to those of natural language tokens, demonstrating that our method systematically improves the statistical structure of audio token sequences. These results highlight the general applicability of our approach, and reinforce the key finding that autoregressive compatibility can be effectively introduced into audio tokenization without sacrificing compression performance.

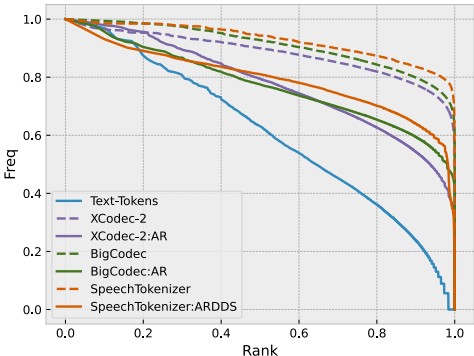

Figure 6: Plot of Zipf's Law comparison between baselines and our method augmented counterparts.

As shown in Figure 7, across downstream tasks (StoryCloze, TopicStoryCloze, AISHELL-I, and SeedTTS), the SpeechLM trained on our augmented codecs consistently outperform their vanilla counterparts. This trend mirrors the results we observed earlier with `XCodec` vs. `XCodec:ARDDS`, confirming that our method improves the tokenization in a way that benefits autoregressive modeling across different codec architectures.

## 5 CONCLUSION

In this work, we revisit the problem of speech codec design from the perspective of autoregressive generative modeling. While existing codecs focus primarily on compression fidelity, their discrete token outputs are not inherently aligned with the autoregressive learning paradigm required by large language models (LLMs). We propose a general training framework that introduces an autoregressive regularization objective and a heterogeneous downsampling strategy to encourage the generation of speech tokens that are both compact and autoregressively coherent. Our method is codec-agnostic and can be applied to a wide range of neural audio tokenizers. Through extensive experiments across multiple codecs, we demonstrate that our approach consistently improves the quality of audio token sequences for SpeechLLM training. Statistical analysis further confirms that our method induces more Zipf-like distributions in the generated audio tokens, aligning them more closely with natural language patterns. We hope this work opens new directions for unifying compression objectives with generative modeling requirements in future audio and multimodal systems.

## ETHICS STATEMENT

This work adheres to the ICLR Code of Ethics. No human subjects or animal experimentation were involved. All datasets used (including LibriSpeech) were obtained in accordance with their respective usage guidelines, ensuring compliance with privacy requirements. No personally identifiable information was used, and the study raises no privacy or security concerns. We have taken care to minimize potential biases and discriminatory outcomes, and are committed to maintaining transparency and integrity throughout the research process.

## REPRODUCIBILITY STATEMENT

We have made our best to ensure the reproducibility of our results. All relevant code and datasets are obey the anonymous policy. Code from prior work used in this study is publicly available for facilitating replication and verification. Details of the experimental setup, including training procedures, model configurations, and hardware specifications, are described in the paper. We believe these measures will enable other researchers to reproduce our findings and build upon them.

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

## A  LLM USAGE

Large Language Models (LLMs) were used to aid in our writing of this work. We use LLMs for checking the grammar of our written manuscript. The prompt is "check the grammar of the following text and give suggestions for making it more academic: ${draft-content-here}".

Some cases list as follows:

**Case 1**:

- **Input:**

check the grammar of the following text and give suggestions for making it more academic: While several codecs (Zeghidour et al., 2021; Kumar et al., 2023; Ye et al., 2025a) apply causal archi-tectures that ensure no information leakage from future tokens into past hidden states, such causal structures do not guarantee autoregressive predictability of the generated speech tokens. This issue is also reflected on the statistical analysis shown on Figure 1.

- **Output:**

Here's a polished version of your text with grammar fixes and suggestions for making it more aca-demic in tone and clarity:

Original: While several codecs (Zeghidour et al., 2021; Kumar et al., 2023; Ye et al., 2025a) apply causal architectures that ensure no information leakage from future tokens into past hidden states, such causal structures do not guarantee autoregressive predictability of the generated speech tokens. This issue is also reflected on the statistical analysis shown on Figure 1.

Revised (academic): Although several codecs (Zeghidour et al., 2021; Kumar et al., 2023; Ye et al., 2025a) adopt causal architectures that prevent information leakage from future tokens into past hid-den states, such architectures do not necessarily guarantee the autoregressive predictability of the generated speech tokens. This limitation is further evidenced by the statistical analysis presented in Figure 1.

Key improvements & rationale:

Replaced "While" with "Although" → smoother for academic writing.

Changed "apply causal architectures" → "adopt causal architectures" (more precise academic verb).

Added "necessarily" → softens the claim and increases nuance.

Changed "reflected on" → "evidenced by" (more formal and accurate).

Changed "shown on Figure" → "presented in Figure" (correct academic phrasing).

Would you like me to make it more concise (single sentence) or keep it two sentences for clarity?

**Case 2**:

- **Input:**

check the grammar of the following text and give suggestions for making it more academic: All speech is tokenized using the trained codecs under evaluation.

- **Output:**

Here's a refined version of your sentence, with grammar corrections and academic tone improvements:

Original: All speech is tokenized using the trained codecs under evaluation.

Revised (academic): All speech samples are tokenized with the trained codecs under evaluation.

Suggestions & rationale:

Changed "speech" → "speech samples" for precision (since you are referring to data instances, not speech as an abstract concept).

Replaced "using" with "with" → slightly more formal and less colloquial.

Kept the passive voice since it fits academic style, but you could also write:

"We tokenize all speech samples with the trained codecs under evaluation." (active, clearer responsibility).

Do you want me to keep the phrasing passive (more impersonal, common in academic writing) or active (clearer attribution, often preferred in CS/ML papers)?

