# OpenReview forum: "Speech Codecs Beyond Compression: Towards Autoregressive Generative Modeling"
_ICLR.cc/2026/Conference — ICLR 2026 Conference Withdrawn Submission_

### Official Review · Reviewer_8kZF · 2025-10-27

**Soundness:** 2
**Presentation:** 2
**Contribution:** 2
**Rating:** 2
**Confidence:** 5

**Summary:**

One of the primary goals of speech codecs is to support language modeling. However, existing codec learning methods only optimize for reconstruction loss or semantic distillation loss. This paper asks: “Why not directly optimize codec learning for autoregressive (AR) language modeling?” The proposed approach is simple yet seemingly effective — they add an additional Transformer to perform AR modeling on the codes during codec learning as a form of regularization. The entire process is trained end-to-end. Their observations (such as the Zipf’s Law analysis comparing text and audio tokens) are interesting, but the major issue lies in the unclear and unconventional evaluation setup for the speech LLM, which raises concerns about the validity of the contribution.

**Strengths:**

- The idea is simple, novel and reasonable
- The analysis on the Zipf’s Law demonstrates that the newly learned codec is more similar to natural language.

**Weaknesses:**

**Major issues:**
- The most serious issue is that the paper describes the codecs as being trained on LibriSpeech (960 hours, English), while the downstream speech LLM is trained on a Chinese audio corpus containing approximately 400k hours, which is not public and whose source is completely unknown. Why? This problematic setting undermines the entire paper. Most of the experimented codecs in the paper are trained for English, and the paper itself further trains on English. Therefore, the evaluation of its language modeling capabilities should be conducted on English. While it is interesting to see the OOD generalizability on Chinese (a nice-to-have), the results on English are required to validate the newly proposed method’s effectiveness. It is entirely possible that, on an English speech LLM, the proposed AR-regularized codec performs almost the same as SpeechTokenizer or XCodec, and the improvement is trivial, given that these existing codecs already have strong language properties due to SSL distillation or reconstruction. English speech LLM evaluation is necessary to gauge the superiority of the AR loss over the prevailing semantic distillation or reconstruction loss. Without the English version comparison, the paper is not complete and should be rejected.

- It is natural to question the reconstruction quality given the added AR regularization and further downsampling for textual length alignment. Although the paper presents evaluations on objective metrics (WER, speaker similarity, and UTMOS), the evaluation is not comprehensive enough—especially due to the lack of standardized MOSHA human evaluation. Furthermore, the demo page is missing.

**Minor issues:**
- The presentation is unclear in several sections.
- If you follow RVQ tokenization for multi-layer tokens, you should cite SoundStream in Section 3.1 and mention RVQ. Otherwise, it is unclear what you mean by multi-level tokenization—it could also mean semantic and acoustic tokens with separate encoders operating at different information levels.
- If you follow RVQ, the first layer is additionally downsampled before quantization. How do you compute the residual for the second-layer quantization? I can probably guess the approach, but it should be explicitly described in the paper or cite a work with a similar approach. It is very unclear in the current form.
- It is also very unclear how the speech LLM is trained. The three cited papers adopt different approaches—for example, Moshi adopts an inner-monologue delayed pattern with the RQ-Transformer, while XCodec adopts a VALL-E-style AR and then NAR hybrid approach. Please elaborate in detail how you train ASR, TTS, and interleaved speech-text tasks with the multi-layer tokens in the Appendix and rebuttal.

**Questions:**

Questions are in the minor issues of the Weakness section.

---

### Official Review · Reviewer_921X · 2025-10-31

**Soundness:** 2
**Presentation:** 3
**Contribution:** 2
**Rating:** 4
**Confidence:** 5

**Summary:**

This paper attempts to enhance speech codec capabilities by introducing two primary techniques: 1) incorporating an autoregressive (AR) regularization loss on the quantized tokens during codec training , and 2) applying heterogeneous downsampling rates to differentiate between semantic and acoustic token layers. The experimental validation is heavily focused on downstream tasks (e.g., ASR, TTS) and includes ablation analyses. While the experimental workload is substantial, the methodology and evaluation exhibit several limitations.

**Strengths:**

The experimental workload is substantial

**Weaknesses:**

1. The introduction of an autoregressive loss imposes a strict autoregressive assumption on the codec's token generation. This conflicts with many modern codec architectures that are non-autoregressive (NAR) or diffusion-based. This AR bias, while beneficial for the specific downstream LLMs, may degrade the tokens' utility for other common tasks (e.g., classification, retrieval) that do not rely on an AR paradigm, thereby limiting the codec's general applicability.
2. The proposed heterogeneous downsampling strategy is contingent on an explicit, multi-layer architecture that separates semantic (0th layer) and acoustic tokens. This approach is not applicable to many common codec designs, such as single-layer codecs (like XCodec-2 or BigCodec, as noted in the paper or models where semantic and acoustic information is fused within a single token stream. Furthermore, this non-uniform structure introduces modeling complexities. Downstream models utilize delay-based patterns and presumably handle padding for the lower-rate semantic layer. This requires the model to distinguish between genuine semantic tokens and special padding tokens, which is a less elegant solution.
3. The core methodological contributions—adding a regularization AR loss and using different sampling rates for different feature layers—are relatively straightforward concepts rather than fundamentally novel approaches to audio tokenization.
4. A more rigorous analysis of the reconstruction trade-off is missing. However, intuitively, imposing an AR constraint may create a trade-off impacting reconstruction fidelity.
5. The baselines were re-trained by the authors. This may introduce performance discrepancies compared to the original, published models.
6. The paper fails to report the final bitrates (e.g., in kbps) for the baseline or proposed codecs
7. There is a significant and unusual data mismatch. The codecs are trained on the English LibriSpeech 585-hour subset, but the downstream SpeechLM is trained on a massive 400k-hour Chinese audio corpus.
8. The 400k-hour Chinese corpus is an internal, non-public dataset. This hinders reproducibility and introduces potential dataset-specific biases.

**Questions:**

Regarding the autoregressive regularization (Section 3.2): The input to the auxiliary AR decoder is the "soft token distribution" $P_t(i)$ derived from Eq. 6. However, the paper does not specify the ground truth (GT) target used to compute the loss $\mathcal{L}_{AR}$ (Eq. 5).

---

### Official Review · Reviewer_pofz · 2025-10-31

**Soundness:** 3
**Presentation:** 3
**Contribution:** 2
**Rating:** 4
**Confidence:** 4

**Summary:**

This paper addresses the critical and timely problem of the mismatch between compression-oriented neural speech codecs and the needs of autoregressive generative models. The authors propose a novel training framework that incorporates two key ideas: an autoregressive regularization loss to encourage token predictability and a heterogeneous downsampling strategy to better align the semantic token rate with that of text. The proposed methods are technically sound, well-motivated, and demonstrated to be highly effective through extensive experiments. The work shows significant improvements in downstream SpeechLM performance across reasoning and generation tasks, without compromising the codec's original compression quality.

**Strengths:**

1. The proposed two-pronged solution is both novel and elegant. The autoregressive regularizer directly optimizes for token sequence predictability, while the heterogeneous downsampling of semantic tokens is a clever approach to bridge the modality gap in terms of sequence length and information rate.
2. The experimental evaluation is a major strength. The authors validate their method (ARDDS-XCodec) against multiple strong baselines on a diverse set of downstream tasks, including speech reasoning and generation. The reported performance gains are substantial and consistent.
3. The authors successfully apply their framework to several other representative codecs (SpeechTokenizer, BigCodec, XCodec-2), showing consistent improvements. This demonstrates that the proposed techniques are not specific to one architecture but represent a general principle for improving codec-LLM compatibility.

**Weaknesses:**

1. The paper sets the regularization weight λ to 1 and the softmax temperature τ to 0.01 after a grid search. However, a more detailed discussion on the sensitivity to these parameters would be beneficial. For instance, exploring the trade-off between reconstruction quality and autoregressive predictability as λ varies would provide deeper insights for practitioners.

2. The method introduces an auxiliary autoregressive decoder that is jointly trained with the codec. While described as "lightweight," it would be helpful to quantify the additional computational cost (e.g., increase in training time) this component introduces. This is a practical consideration for researchers aiming to train new codecs using this framework.

3. The authors note that reducing the semantic token rate below 6.25Hz led to training instability. This is an important finding and a potential limitation. A more in-depth discussion or hypothesis about the cause of this instability (e.g., information bottleneck becoming too severe, misalignment with acoustic details) would strengthen the paper.

4. The codecs are trained on the LibriSpeech 960-hour corpus. While this is a standard benchmark, it is relatively modest in the era of web-scale data. The downstream SpeechLM is trained on a much larger 400k-hour corpus, but the fundamental properties of the audio tokenizer (the codec) are learned on the smaller dataset. Evaluating the proposed framework on codecs trained with significantly larger and more diverse data would be a crucial test of its scalability and general effectiveness.

**Questions:**

As noted in the weaknesses section

---

### Official Review · Reviewer_mWBg · 2025-11-03

**Soundness:** 2
**Presentation:** 3
**Contribution:** 2
**Rating:** 2
**Confidence:** 4

**Summary:**

The authors argue that existing speech codecs are optimized for compression rather than compatibility with autoregressive generative models. They propose a novel framework that introduces autoregressive regularization and heterogeneous downsampling strategy (DDS) to align speech tokenization with the temporal consistency and predictability requirements of large language models (LLMs).

The method is evaluated through downstream SpeechLM tasks including reasoning, ASR, and TTS, demonstrating improvements while maintaining codec compression quality.

**Strengths:**

1. The manuscript indicates an interesting but overlooked issue, that compression-oriented codecs don't naturally align with autoregressive modelling paradigms. This perspective is insightful and well-motivated.

2. The autoregressive regularizer idea is conceptually simple, easy to integrate with existing codecs, and differentiable through soft assignment. The heterogeneous downsampling idea is intuitive and improves cross-modal alignment.

3. Zipfian analysis is novel and insightful.

**Weaknesses:**

1. The authors report that downsampling the semantic layer below 6.25Hz ed to "training instability". Their hypothesis is that this is because the audio token rate falls below the text token rate. The assumption is interesting but not fully explored. This suggests a potential limitation of the approach that could be investigated further.

2. The paper lacks theoretical analysis of why autoregressive regularization improves token predictability. The connection between Zipfian distributions and autoregressive modeling performance could be formalized more rigorously.

3. The codecs are trained on English (LibriSpeech) , but the downstream SpeechLM is trained on Chinese corpus and evaluated on several Chinese tasks. While it may suggest the method is language-agnostic, this cross-lingual setup is a potential confounding and weird variable that is not discussed in the text.

4. The experiments are limited. The method should be compared with more SOTA codecs and continuous speech representations.

5. Lacking ablations such as on the number of layers of  the auxiliary AR model.

**Questions:**

1. Please refer to the "Weaknesses" part and address the concerns.

2. The LM model is trained on Chinese corpus, and test on SeedTTS-Eval. But as far as I know SeedTTS-Eval set it self both have English and Chinese subset. So which part did you use? Please specify it.

3. What is the computational overhead of adding the autoregressive decoder during codec training? How does training time compare to vanilla codecs? Would a much smaller or larger AR decoder change results?

4. Can you provide theoretical analysis or intuition for why enforcing autoregressive compatibility during codec training improves downstream LLM performance?

5. Can you provide more insight into why the Zipf-like behavior emerges? Is it merely due to increased predictability or does the AR loss explicitly induce long-tail distributions?

---

### Note · Authors · 2025-11-15

I have read and agree with the venue's withdrawal policy on behalf of myself and my co-authors.